# Site-directed Mutagenesis of a β-Glycoside Hydrolase from *Lentinula edodes*

**DOI:** 10.3390/molecules24010059

**Published:** 2018-12-24

**Authors:** Jing-Jing Chen, Xiao Liang, Tian-Jiao Chen, Jin-Ling Yang, Ping Zhu

**Affiliations:** State Key Laboratory of Bioactive Substance and Function of Natural Medicines & NHC Key Laboratory of Biosynthesis of Natural Products, Institute of Materia Medica, Chinese Academy of Medical Sciences & Peking Union Medical College, 1 Xian Nong Tan Street, Beijing 100050, China; chenjingjing@imm.ac.cn (J.-J.C.); liangxiao@imm.ac.cn (X.L.); chentianjiao@imm.ac.cn (T.-J.C.); yangjl@imm.ac.cn (J.-L.Y.)

**Keywords:** site-directed mutagenesis, β-xylosidase, β-glucosidase, catalytic efficiency, molecular docking

## Abstract

The β-glycoside hydrolases (LXYL-P1−1 and LXYL-P1−2) from *Lentinula edodes* (strain M95.33) can specifically hydrolyze 7-β-xylosyl-10-deacetyltaxol (XDT) to form 10-deacetyltaxol for the semi-synthesis of Taxol. Our previous study showed that both the I368T mutation in LXYL-P1−1 and the T368E mutation in LXYL-P1−2 could increase the enzyme activity, which prompted us to investigate the effect of the I368E mutation on LXYL-P1−1 activity. In this study, the β-xylosidase and β-glucosidase activities of LXYL-P1−1^I368E^ were 1.5 and 2.2 times higher than those of LXYL-P1−1. Most importantly, combination of I368E and V91S exerted the cumulative effects on the improvement of the enzyme activities and catalytic efficiency. The β-xylosidase and β-glucosidase activities of the double mutant LXYL-P1−1^V91S/I368E^ were 3.2 and 1.7-fold higher than those of LXYL-P1−1^I368E^. Similarly, the catalytic efficiency of LXYL-P1−1^V91S/I368E^ on 7-β-xylosyl-10-deacetyltaxol was 1.8-fold higher than that of LXYL-P1−1^I368E^ due to the dramatic increase in the substrate affinity. Molecular docking results suggest that the V91S and I368E mutation might positively promote the interaction between enzyme and substrate through altering the loop conformation near XDT and increasing the hydrogen bonds among Ser^91^, Trp^301^, and XDT. This study lays the foundation for exploring the relationship between the structure and function of the β-glycoside hydrolases.

## 1. Introduction

As the biocatalysts, enzymes are widely used in the production of food products, commodities, and pharmaceutical intermediates [1,2]. The prompt developments in protein engineering technology have provided the useful tools for improving enzyme critical traits, such as stability and catalytic efficiency [3,4,5,6,7,8]. The common strategies for protein engineering include directed evolution and rational protein design [9,10]. Directed evolution is a method that mimics the natural evolution in the laboratory. It utilizes the error-prone PCR or DNA shuffling technique in combination with the high-throughput screening method to continuously accumulate the dominant mutations with improved characteristics of the enzyme [5,11,12,13,14,15]. Rational design is conducted based on the understanding of the catalytic mechanism or the enzyme structure in which the stereo-structure can sometimes be predicted by protein homology modeling technique [16,17,18,19,20,21]. The key amino acids that may affect the enzyme properties can be chosen for site-directed mutagenesis, which includes the single site-directed mutation, multiple site-directed mutations, and saturation mutation. For example, the thermostability of *Geobacillus stearothermophilus* xylanase was improved by directed evolution in combination with rational design and up to 13 amino acids were mutated during this process. The reaction temperature for maximum activity increased from 77 °C to 87 °C, and the catalytic efficiency increased by 90% [13]. Through DNA shuffling, site-directed mutation and saturated mutation, the stabilities and activities of the β-glucosidases from *Thermobifida fusca* and *Paebibacillus polymxyxa* were significantly increased, making the enzymes more suitable for the bioconversion of cellulose [22]. By site-directed mutation of three His (His^275^, His^293^, and His^310^) of the α-amylase in *Bacillus subtilis* into Asp, the catalytic efficiency of the mutant on the substrate was improved by 16.7 times compared with that of the wild type [23]. Additionally, the mutations of P140L/D416G significantly increased the catalytic efficiency of the mannanase from *Podospora anserina* [24]. All of the aforementioned examples suggested that protein engineering can promote the study of the enzyme structure-function relationship and can be used to design enzymes with improved or new functions, which will broaden the repertoire of enzymes.

The dual GH3 β-xylosidase/β-glucosidases, designated as LXYL-P1−1 and LXYL-P1−2, respectively, are enzymes of *Lentinula edodes* (strain M95.33) origin and have been cloned and characterized by our lab. The activity of LXYL-P1−2 is twice higher than that of LXYL-P1−1 [25]. In addition, both enzymes have been successfully expressed in *Pichia pastoris*. Moreover, both of them can specifically remove the xylosyl group from 7-xylosyl-10-deacetyltaxol (XDT) isolated from yew trees to produce 10-deacetyltaxol (DT) [25,26,27]. This product can be further acetylated into Taxol, a prominent anticancer drug originally isolated from Pacific yew tree [16,28,29,30,31]. In our previous study, the directed evolution of LXYL-P1−2 had been conducted. From the random mutant library created by error-prone PCR, we obtained a mutant LXYL-P1−2-EP2 (LXYL-P1−2^T368E^) that harbored the T368E mutation, which exhibited a 47% increase in its catalytic efficiency on XDT and elevated stability in the range of pH ≥ 6 compared with LXYL-P1−2 [32]. Recently, we found that the mutant LXYL-P1−1^I368T^ also exhibited similar β-xylosidase and β-glucosidase activities compared with the high-active LXYL-P1−2, although the activities were lower than those of LXYL-P1−1^A72T^, LXYL-P1−1^V91S^ (the most active single mutant), and the double mutant LXYL-P1−1^A72T^^/V91S^ [33]. These results suggest that besides the putative active site residues [21] and in addition to positions 72 and 91, the amino acid in position 368 may also play an important role in terms of enzyme activity. In addition, we also observed that the double mutant LXYL-P1−1^A72T^^/V91S^ even showed results 2.8- and 3-fold higher than the positive control LXYL-P1−2 on β-xylosidase and β-glucosidase activities, although the triple mutant LXYL-P1−1^A72T/V91S/I368T^ did not exhibit increased activities compared with the same control [33]. These results prompted us to consider whether the I368E mutation in LXYL-P1−1 can also increase enzyme activities and whether the enzyme activities can be further improved by the combination mutation of V91S/I368E. In this study, the single mutant LXYL-P1−1^I368E^ and the double mutant LXYL-P1−1^V91S/I368E^ as well as their corresponding engineered yeasts were constructed, respectively. With respect to the results, we discovered that the mutant LXYL-P1−1^I368E^ was indeed more active than LXYL-P1−1. Furthermore, the double mutant LXYL-P1−1^V91S/I368E^ even surpassed the high-active LXYL-P1−2 in terms of the β-xylosidase and β-glucosidase activities. The possible mechanisms are further discussed.

## 2. Results and Discussion

### 2.1. The Volumetric and Biomass Enzyme Activities of the Recombinant Yeasts

In order to investigate the effect of I368E mutation on the enzyme activity, the recombinant yeast GS115-3.5K-P1−1^I368E^ was constructed, and its volumetric and biomass enzyme activities were detected as described previously [32]. After induction by methanol for 4 days, the enzyme activities of GS115-3.5K-P1−1^I368E^ had exceeded those of GS115-3.5K-P1−1 (Figure 1). At the induction time of 7 d, the volumetric and biomass β-xylosidase activities of GS115-3.5K-P1−1^I368E^ reached 3.52 × 10^6^ U·L^−1^ and 0.72 × 10^5^ U·g^−1^, respectively, which increased by 21% and 18% compared with those of GS115-3.5K-P1−1 (2.92 × 10^6^ U·L^−1^ and 0.61 × 10^5^ U·g^−1^, respectively) (Figure 1a,b). Similarly, at the induction time of 7 days, the volumetric and the biomass β-glucosidase activities of GS115-3.5K-P1−1^I368E^ arrived at 6.34 × 10^6^ U·L^−1^ and 1.30 × 10^5^ U·g^−1^, respectively, which increased by 23% and 21% compared with those of GS115-3.5K-P1−1 (5.14 × 10^6^ U·L^−1^ and 1.07 × 10^5^ U·g^−1^, respectively) (Figure 1c,d).

In our previous study, we found that the A72T mutation and V91S mutation exhibited a synergistic effect, in which the amino acid in position 91 of LXYL-P1−1 displayed a key role in affecting enzyme activity [33]. This synergistic effect was also observed in the double mutant V91S/I368E as shown in Figure 1. The enzyme activities of GS115-3.5K-P1−1^V91S/I368E^ exceeded those of the control strain GS115-3.5K-P1−1^I368E^ during the whole methanol induction period. Its volumetric and biomass β-xylosidase activities reached 11.51 × 10^6^ U·L^−1^ and 2.32 × 10^5^ U·g^−1^, respectively, at the induction time of 7 days, which were 3.3- and 3.2-fold higher than those of GS115-3.5K-P1−1^I368E^ (Figure 1a,b). Likewise, the volumetric and biomass β-glucosidase activities of GS115-3.5K-P1−1^V91S/I368E^ reached 20.67 × 10^6^ U·L^−1^ and 4.18 × 10^5^ U·g^−1^, respectively, at the induction time of 7 days, which were 3.3- and 3.2-fold higher than those of GS115-3.5K-P1−1^I368E^ (Figure 1c,d).

### 2.2. Specific β-Xylosidase and β-Glucosidase Activities of the Mutants

The specific activities of the purified mutants were also detected. As shown in Figure 2, the β-xylosidase and β-glucosidase activities of LXYL-P1−1^I368E^ reached 3.41 × 10^4^ and 10.80 × 10^4^ U/mg, respectively, which were 1.5 and 2.2 times as high as those of LXYL-P1−1 (2.33 × 10^4^ and 4.93 × 10^4^ U/mg, respectively), although the activities were lower than those of LXYL-P1−1^I368^^T^ reported previously [33]. The β-xylosidase and β-glucosidase activities of the purified LXYL-P1−1^V91S/I368E^ reached 11.04 × 10^4^ and 18.27 × 10^4^ U/mg, respectively, which were 4.7 and 3.7 times higher than those of LXYL-P1−1, and 3.2 and 1.7-fold higher than those of LXYL-P1−1^I368E^, and even 2.3- and 1.5-fold higher than those of LXYL-P1−2 (4.80 × 10^4^ and 11.85 × 10^4^ U/mg, respectively) (Figure 2). The results indicate that the I368E mutation in LXYL-P1−1 presented here has exhibited a positive effect on increasing the β-xylosidase and β-glucosidase activities. Meanwhile, the combination of V91S and I368E mutations had a synergetic effect on the increase of the β-xylosidase and β-glucosidase activities. Further, compared with the volumetric or biomass enzyme activity of the recombinant yeast represented in Figure 1, we found that the increased magnitude of the specific activity of LXYL-P1-1^I368E^ was apparently higher than that of the volumetric or biomass activity of GS115-3.5K-LXYL-P1-1^I368E^. It means that the single mutation led to the decreased enzyme expression in the yeast host.

### 2.3. Kinetic Analysis of the Mutated Enzymes against XDT

The kinetic parameters of LXYL-P1−1^I368E^ and LXYL-P1−1^V91S/I368E^ together with LXYL-P1−1^I368^^T^ against XDT were detected and the data are summarized in Table 1. To LXYL-P1−1^I368^^T^, the increased affinity (*K*_m_ value: 0.35 vs. 0.50, mM) contributed to the slight improvement in its catalytic efficiency compared with that of LXYL-P1−1 (*k*_cat_/*K*_m_ value: 4.46 vs. 4.10, s^−1^·mM^−1^). To LXYL-P1−1^I368E^, both the increased affinity (*K*_m_ value: 0.42 vs. 0.50, mM) and the increased turnover number (*k*_cat_ value: 2.41 vs. 2.05, s^−1^) led to the significant increase in its catalytic efficiency compared with that of LXYL-P1−1 (*k*_cat_/*K*_m_ value: 5.68 vs. 4.10, s^−1^·mM^−1^) (Table 1). Likewise, the significantly increased affinity (*K*_m_ value: 0.20 vs. 0.50, mM) and a similar turnover number (*k*_cat_ value: 2.06 vs. 2.05, s^−1^) resulted in the 2.5-fold increase in the catalytic efficiency of LXYL-P1−1^V91S/I368E^ compared with that of LXYL-P1−1 (*k*_cat_/*K*_m_ value: 10.30 vs. 4.10, s^−1^·mM^−1^). In other words, the catalytic efficiency of LXYL-P1−1^I368E^ against XDT was 1.3-fold higer than that of LXYL-P1−1^I368^^T^, and the catalytic efficiency of LXYL-P1−1^V91S/I368E^ was nearly twice as high as that of LXYL-P1−1^I368E^ (Table 1), and also surpassed that of LXYL-P1−1^V91S^ (6.26 s^−1^·mM^−1^) [33].

### 2.4. Substrate-Enzyme Molecular Docking

To further explore how these mutations affect enzyme activities, molecular docking between the mutants and the substrate XDT was conducted based on the virtual three-dimensional structure of LXYL-P1−1, which was previously predicted through molecular modeling homology [33]. As shown in Figure 3a,b, the 368th amino acid is located on the loop and at the surface of the predicted protein. The I368E mutation provided an opportunity to introduce geometrical alteration of the loop near the active pocket, which may lead to enhanced affinity to the substrate. In addition, the I368E substitution gave rise to a negative potential on the protein surface, which probably made the mutant more stable in such a micro-environment. As Ile^368^ is a nonpolar and hydrophobic amino acid and Glu^368^ is a polar and acidic amino acid, it is likely that the introduction of a polar residue in position 368 may contribute to enzyme stability, and had an important effect on improving enzyme activity. Moreover, the previous study suggests that the V91S might increase the hydrogen bonds among Ser^91^, Trp^301^, and XDT [33]. This phenomenon may also occur in the present study (Figure 3c,d), since the remarkably increased affinity of the double mutant LXYL-P1−1^V91S/I368E^ (*K*_m_ value: 0.12 mM) to the substrate XDT was observed (Table 1). 

## 3. Materials and Methods 

### 3.1. Plasmids and Strains

The recombinant plasmid pPIC3.5K-LXYL-P1−1 harboring the *lxyl-p1−1* gene from *L. edodes* M95.33 was previously constructed in our laboratory. *Pichia pastoris* GS115-3.5K-P1−1 was constructed by transforming the plasmid pPIC3.5K-LXYL-P1−1 into the host strain *P. pastoris* GS115 (Mut^+^), and preserved at −80 °C prior to use [25].

### 3.2. Construction of the Recombinant Plasmids Expressed lxyl-p1−1^I368E^ and lxyl-p1−1^V91S/I368E^

The *lxyl-p1−1* variants harboring single site-directed mutation or double site-directed mutations were amplified using the PCR-based overlap extension method. The primers used for the amplification are listed in Table 2. For the construction of *lxyl-p1−1*^I368E^, the two individual fragments were amplified by Phusion DNA polymerase using primers P1−1-F/I368E-R and I368E-F/P1−1-R, respectively, with the plasmid pPIC3.5K-LXYL-P1−1 being used as a template. The PCR conditions for amplification consisted of 98 °C for 30 s, 30 cycles of 10 s at 98 °C, 30 s at 60 °C, 1 min at 72 °C, and a final 10 min extension at 72 °C. The PCR products were purified using a gel extraction kit (Transgen, Beijing, China). Later, the overlap extension was performed by mixing 100 ng of the two fragments in equimolar amounts with Phusion PCR buffer, dNTPs, and Phusion polymerase in a total volume of 25 μL. The PCR conditions for amplification were 98 °C for 30 s, 15 cycles of 10 s at 98 °C, 30 s at 60 °C, 72 °C for 30 s/kb, followed 10 min incubation at 72 °C. Then, 2 μL of the unpurified PCR product was further used as a template for the second round PCR. Additionally, P1−1-F and P1−1-R, Phusion PCR buffer, dNTPs, and Phusion polymerase were added into the PCR mixture in a final volume of 50 μL. The amplification was performed identically to the PCR reaction of the individual fragments. Finally, the fragment *lxyl-p1−1*^I368E^ containing the I368E mutation was obtained. For the construction of *lxyl-p1−1*^V91S/I368E^, the plasmid pPIC3.5K-LXYL-P1−1 was also used as a template, and the three individual fragments were amplified using primers SP1−1-F/V91S-R, V91S-F/I368E-R, and I368E-F/P1−1-R, respectively. Next, the three independent fragments were fused by overlap extension PCR to gain *lxyl-p1−1*^V91S/I368E^. Finally, *lxyl-p1−1*^I368E^ and *lxyl-p1−1*^V91S/I368E^ were ligated into the expression vector pPIC3.5K at the *Bam*H I and *Not* I restriction sites to generate the expression plasmids pPIC3.5K-LXYL-P1−1^I368E^ and pPIC3.5K-LXYL-P1−1^V91S/I368E^, respectively. The recombinant plasmids with site-directed mutations were confirmed by nucleotide sequence analysis. 

### 3.3. Construction of the Recombinant Yeast Expressed lxyl-p1−1^I368E^ and lxyl-p1−1^V91S/I368E^

For construction of engineered *P. pastoris* strains containing multi-copy *lxyl-p1−1*^I368E^ and *lxyl-p1−1*^V91S/I368E^, 10 μg of recombinant vectors (pPIC3.5K-LXYL-P1−1^I368E^ and pPIC3.5K-LXYL-P1−1^V91S/I368E^) were linearized with *Sac* I and introduced into *P. pastoris* GS115 via electroporation transformation according to the manufacturer’s instructions (Invitrogen, Carlsbad, CA, USA). The transformants were initially selected on MD plates (13.4 g/L yeast nitrogen base, 0.4 mg/L biotin, 20 g/L dextrose, and 15 g/L agar) and then screened for multiple integrants on YPD plates (10 g/L yeast extract, 20 g/L tryptone, 20 g/L d-glucose, and 15 g/L agar) containing 4 mg/mL G418. Genomic DNA of the transformants was extracted via TIANamp Yeast DNA Kit following the manufacturer’s instruction, and used for the further PCR analysis.

### 3.4. Induction Protein Expression of LXYL-P1−1^I368E^ and LXYL-P1−1^V91S/I368E^

The recombinant yeasts harboring the *lxyl-p1−1*^I368E^ and *lxyl-p1−1*^V91S/I368E^ were firstly inoculated in 500-mL shake flasks containing 100 mL buffered minimal glycerol complex medium (BMGY) medium (containing 10 g/L yeast extract, 20 g/L tryptone, 13.4 g/L YNB, 0.4 mg/L biotin, 10 g/L glycerol, 100 mmol·L^−1^ potassium phosphate buffer, pH 6.0) at 30 °C with shaking at 200 rpm for 48–60 h. Then methanol was added every day to maintain 1% (*v*/*v*) for the induction of the gene expression. 

### 3.5. Volumetric and Biomass Enzyme Activities Measurement of the Recombinant Yeasts

At the methanol induction stage, the volumetric and biomass β-xylosidase and β-glucosidase activities of the recombinant yeasts were measured every day. The culture was harvested via centrifugation and was washed twice with dH_2_O, and the cell pellet was resuspended with dH_2_O in the same volume of the culture broth. Next, 10 μL of the cell suspension was added to 50 μL of 5 mmol·L^−1^ PNP-Xyl or PNP-Glu, and incubated for 20 min at 50 °C for the catalytic activity analysis. The volumetric and biomass β-xylosidase and β-glucosidase activities were then evaluated as described previously [32].

### 3.6. Enzyme Activities Measurement of LXYL-P1−1^I368E^ and LXYL-P1−1^V91S/I368E^

After 7 days of induction, the recombinant mutants were isolated and purified according to the method described in our previous report [25,32]. The β-xylosidase and β-glucosidase activities of mutants were measured by detecting the amount of p-nitrophenol released from the substrate PNP-Xyl or PNP-Glu under the optimum reaction conditions. Next, 60 μL reaction volume contained 50 μL of 5 mmol·L^−1^ PNP-Xyl/PNP-Glc and 10 μL of diluted enzyme in 50 mmol·L^−1^ sodium acetate buffer with pH 5.0. The reaction was performed under 50 °C for 20 min. Reactions were terminated by adding 1 mL saturated Na_2_B_4_O_7_ solution. The enzymatic activity was assayed using spectrophotometry based on the absorbance at 405 nm. One unit of activity was defined as the amount of enzyme that catalyzed the formation of 1 nmol·L^−1^ p-nitrophenol per minute.

### 3.7. Kinetic Study of LXYL-P1−1^I368E^ and LXYL-P1−1^V91S/I368E^

The kinetic parameters of LXYL-P1−1^I368E^ and LXYL-P1−1^V91S/I368E^ against XDT were determined at the XDT concentration ranging from 0.039–5.0 mmol·L^−1^ as described previously [32]. DT formation was analyzed through HPLC. The kinetic data on XDT were processed by a proportional weighted fit using a nonlinear regression analysis program based on Michaelis–Menten enzyme kinetics. All data were presented as means ± SD of three independent repeats.

## 4. Conclusions and Perspective

In conclusion, the site-directed mutagenesis of the amino acid in position 368 of LXYL-P1−1 was conducted, and the mutant with the I368E mutation had exhibited increased β-xylosidase and β-glucosidase activities. Moreover, combination of I368E and V91S could further significantly improve the enzyme activity and catalytic efficiency. The increased catalytic efficiency of LXYL-P1−1^V91S/I368E^ on XDT was mainly due to the dramatic increase in the substrate affinity. Molecular docking analysis between the mutants and XDT deduced the possible molecular mechanism for the improved enzyme activities. Our results suggest that combination of two or more beneficial mutations should probably improve the enzyme activities. In the future, the saturation mutation on the 368th site of LXYL-P1−1 followed by the other combinatorial mutations (including A72T/I368E, A72T/I368T and V91S/I368T) may be conducted to find more active mutants. The corresponding high-active mutant can be further used for the bioconversion of XDT to DT for the semi-synthesis of Taxol. This study provides the theoretical basis for the identification of the important key amino acid residues out of active sites that positively affect the activities of the β-glycoside hydrolases, and lays the foundation for further exploring the relationship between the structure and function of the β-glycoside hydrolases.

## Figures and Tables

**Figure 1 molecules-24-00059-f001:**
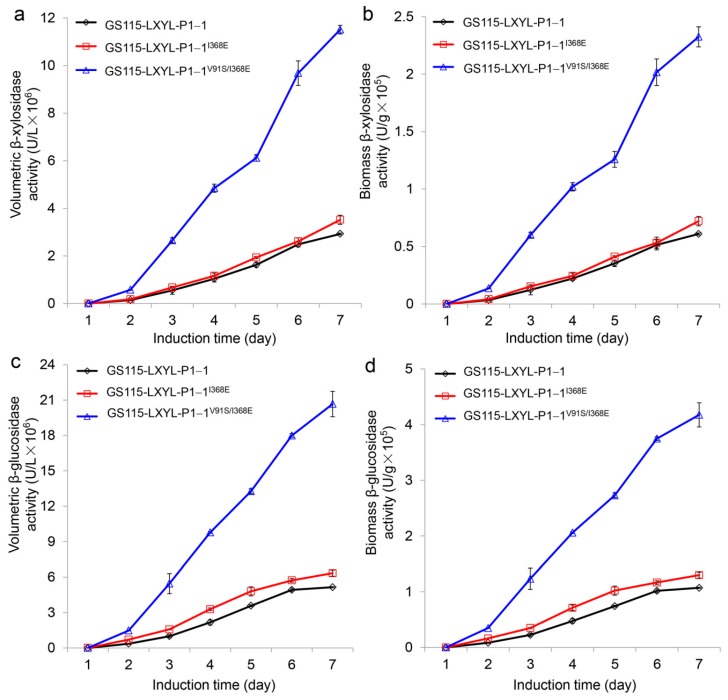
Comparison of β-xylosidase and β-glucosidase activities between the recombinant yeasts GS115-3.5K-LXYL-P1−1, GS115-3.5K-LXYL-P1−1^I368E^, and GS115-3.5K-LXYL-P1−1^V91S/I368E^. The recombinant yeast GS115-3.5K-LXYL-P1−1 was used as the control. (**a**) Volumetric β-xylosidase activities. (**b**) Biomass β-xylosidase activities. (**c**) Volumetric β-glucosidase activities. (**d**) Biomass β-glucosidase activities.

**Figure 2 molecules-24-00059-f002:**
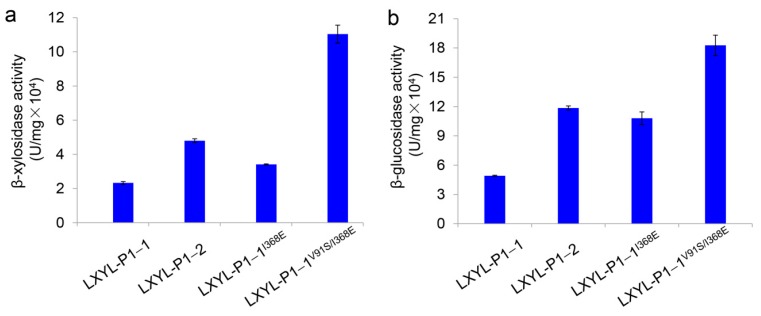
Comparison of β-xylosidase (**a**) and β-glucosidase (**b**) activities between the purified mutants. Data are mean ± SD. *n* = 3.

**Figure 3 molecules-24-00059-f003:**
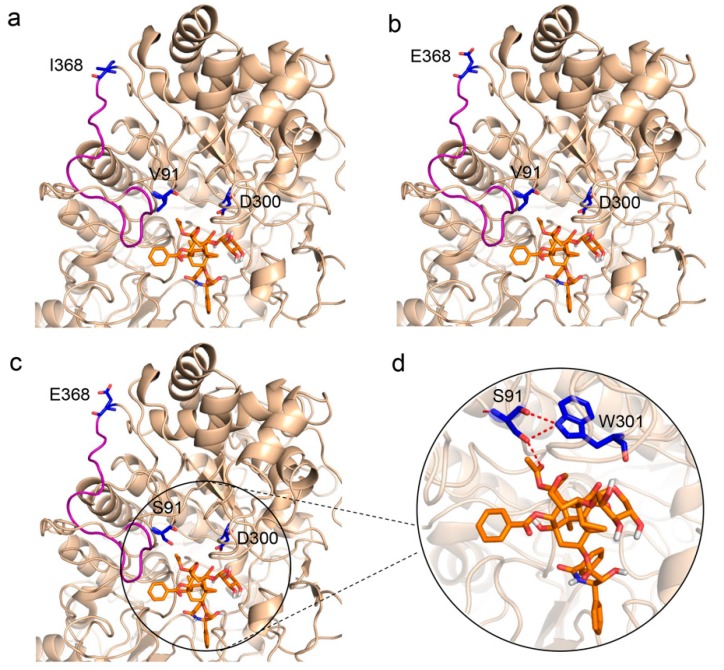
Partial view of enzyme-XDT docking. (**a**) Side view of LXYL-P1−1 with XDT, showing Val^91^ and Ile^368^. (**b**) Side view of LXYL-P1−1^I368E^ with XDT, in which Ile^368^ are replaced by Glu^368^. (**c**) Side view of LXYL-P1−1^V91S/I368E^ with XDT, in which Val^91^ and Ile^368^ are replaced by Ser^91^ and Glu^368^, respectively. (**d**) Enlarged view of molecular docking of LXYL-P1−1^V91S/I368E^ with XDT, in which the increased hydrogen bonds among Ser^91^, Trp^301^, and XDT are indicated in red. The geometrical alteration of the loop near the active pocket is indicated in salmon. The carbon atoms of XDT are shown in orange. The nucleophile Asp^300^ (catalytic site), Val^91^/Ser^91^ and Ile^368^/Glu^368^ are colored in blue.

**Table 1 molecules-24-00059-t001:** Kinetic parameters for the mutated enzymes using XDT as the substrate.

	*V*_max_ (μM·min^−1^)	*K*_m_ (mM)	*k*_cat_ (s^−1^)	*k*_cat_/*K*_m_ (s^−1^·mM^−1^)
LXYL-P1−1	3.42 (±0.04)	0.50 (±0.01)	2.05 (±0.02)	4.10
LXYL-P1−1^I368T^	2.60 (±0.56) **	0.35 (±0.10) *	1.56 (±0.34) *	4.46 *
LXYL-P1−1^I368E^	4.02 (±0.13) *	0.42 (±0.01) *	2.41 (±0.08) *	5.74 ***
LXYL-P1−1^V91S/I368E^	3.43 (±0.01)	0.20 (±0.01) ** ^##^	2.06 (±0.01)	10.30 *** ^###^

Note: Data are mean (±SD), *n* = 3. * *p* < 0.05 vs. LXYL-P1−1, ** *p* < 0.01 vs. LXYL-P1−1, *** *p* < 0.01 vs. LXYL-P1−1; ^##^
*p* < 0.01 vs. LXYL-P1−1^I368E^, ^###^
*p* < 0.001 vs. LXYL-P1−1^I368E^.

**Table 2 molecules-24-00059-t002:** The primers used for amplification of *lxyl-p1−1* variants.

Primer	Sequence (5′→3′)
P1−1-F	CGCGGATCCATGTTCTCAGCAAGAC
P1−1-R	TTTTCCTTTTGCGGCCGCTCAGTGGTGGTGGTGG
V91S-F	GAATTAGCCAACATCACCTCAGGGGTTATAGGTTTGTGT$\begin{array}{c} \mathrm{TCA} \end{array}$GGAGTA
V91S-R	TACTCC$\begin{array}{c} \mathrm{TGA} \end{array}$ACACAAACCTATAACCCCTGAGGTGATGTTGGCTAATTC
I368E-F	CAAGAT$\begin{array}{c} \mathrm{GAA} \end{array}$AATCCACCACCACCCTTTG
I368E-R	TGGATT$\begin{array}{c} \mathrm{TTC} \end{array}$ATCTTGACCGAGGTAATAG

Note: The underlined is the restriction enzyme cleavage site. The mutated bases are indicated in box.

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
