# Peer review of "Site-directed Mutagenesis of a β-Glycoside Hydrolase from Lentinula edodes"

_molecules, 2018, doi:10.3390/molecules24010059_

Reviewer 1 Report

This manuscript describes engineered glycoside hydrolases from Lentinula edodes for improved conversion of XDT to DT on the synthetic path to Taxol, a key pharmaceutical. The work and findings are therefore significant.

In addition to minor English language corrections (see attached "Molecules 2018 Manuscript Review" pd file), here are specific questions and suggested corrections (note that these also appear in the attached pdf file):

Suggested modifications/clarifications

Did you consider combining Figures 1 and 2 to ease comparison among all of the recombinants? As currently set up, you have the data for GS115-LXYL-P1‒1I368E presented twice. At the least, please consider changing the series (symbol type/color and line color) for GS115-LXYL-P1‒1I368E to match in the two Figures.

Line 19  See the comment at Line 135; presuming that kcat/Km for LXYL-P1‒1V91S/I368E should be 17.17 mM-1 s-1, that’s a 3.0-fold increase in catalytic efficiency relative to kcat/Km for LXYL-P1‒1I368E of 5.74 mM-1 s-1.

Line 109  The legend isn’t consistent with the caption in Figure 2. In the caption, do you mean to say GS115-LXYL-P1‒1I368E rather than GS115-LXYL-P1‒1? If yes, what is the control,
GS115-LXYL-P1‒1I368E or GS115-LXYL-P1‒1? According to the text that should be
GS115-LXYL-P1‒1I368E.

Line 120  What are the specific values for the LXYL-P1‒2 xylosidase and glucosidase activities?

Line 125  What do the error bars represent in Figure 3?

Line 131  I calculate kcat/Km for LXYL-P1‒1I368E of 5.74 mM-1 s-1 (2.41 s-1/0.42 mM). Correction would also need to be made in Table 1.

Line 133  I calculate an increase in kcat/Km for LXYL-P1‒1V91S/I368E relative to LXYL-P1‒1 of 4.2 (see next line edit).

Line 134  I calculate kcat/Km for LXYL-P1‒1V91S/I368E of 17.17 mM-1 s-1 (2.06 s-1/0.12 mM). Correction would also need to be made in Table 1.

Line 135  Presuming that kcat/Km for LXYL-P1‒1V91S/I368E should be 17.17 mM-1 s-1, that’s a 3.0-fold increase in catalytic efficiency relative to kcat/Km for LXYL-P1‒1I368E of 5.74 mM-1 s-1.

Line 174  As for Line 135, that should be a 3.0-fold higher catalytic efficiency.

Line 181 Have you considered the LXYL-P1‒1A72T mutation?

Line 191  Transformation with what? Should that “with” be there?

Author Response

Response to Reviewer 1 Comments

Point 1: Did you consider combining Figures 1 and 2 to ease comparison among all of the recombinants? As currently set up, you have the data for GS115-LXYL-P1‒1I368E presented twice. At the least, please consider changing the series (symbol type/color and line color) for GS115-LXYL-P1‒1I368E to match in the two Figures.

Response 1: Thank you for your suggestion. We have combined Figures 1 and 2 in our revised manuscript.

Point 2: Line 19 See the comment at Line 135; presuming that kcat/Km for LXYL-P1‒1V91S/I368E should be 17.17 mM-1 s-1, that’s a 3.0-fold increase in catalytic efficiency relative to kcat/Km for LXYL-P1‒1I368E of 5.74 mM-1 s-1.

Response 2: Sorry for this error. The Km value for LXYL-P1‒1V91S/I368E is 0.20 mM, not 0.12 mM. Thus, the kcat/Km for LXYL-P1‒1V91S/I368E is still 10.30 mM-1 s-1. We have corrected the Km value for LXYL-P1‒1V91S/I368E in the revised manuscript.

Point 3: Line 109  The legend isn’t consistent with the caption in Figure 2. In the caption, do you mean to say GS115-LXYL-P1‒1I368E rather than GS115-LXYL-P1‒1? If yes, what is the control, GS115-LXYL-P1‒1I368E or GS115-LXYL-P1‒1? According to the text that should be GS115-LXYL-P1‒1I368E.

Response 3: Thank you for pointing out the error. The recombinant yeast GS115-3.5K-LXYL-P1-1I368E was used as the control in original Figure 2. Now, Figures 1 and 2 have been combined and the error has been corrected.

Point 4: Line 120  What are the specific values for the LXYL-P1‒2 xylosidase and glucosidase activities?

Response 4: The specific values for the LXYL-P1‒2 xylosidase and glucosidase activities were 4.8×104 and 11.85×104 U/mg, respectively. We have added the values in the revised manuscript.

Point 5: Line 125  What do the error bars represent in Figure 3?

Response 5: The error bars represent that data are mean ± SD. n=3. We have added this in the legend of Figure 3.

Point 6: Line 131  I calculate kcat/Km for LXYL-P1‒1I368E of 5.74 mM-1 s-1 (2.41 s-1/0.42 mM). Correction would also need to be made in Table 1.

Response 6: We have revised this value in the revised manuscript and Table 1.

Point 7: Line 133  I calculate an increase in kcat/Km for LXYL-P1‒1V91S/I368E relative to LXYL-P1‒1 of 4.2 (see next line edit).

Line 134  I calculate kcat/Km for LXYL-P1‒1V91S/I368E of 17.17 mM-1 s-1 (2.06 s-1/0.12 mM). Correction would also need to be made in Table 1.

Line 135  Presuming that kcat/Km for LXYL-P1‒1V91S/I368E should be 17.17 mM-1 s-1, that’s a 3.0-fold increase in catalytic efficiency relative to kcat/Km for LXYL-P1‒1I368E of 5.74 mM-1 s-1.

Line 174  As for Line 135, that should be a 3.0-fold higher catalytic efficiency. 

Response 7: Sorry for the clerical error concerning the Km value for LXYL-P1‒1V91S/I368E in our original manuscript. In fact, the Km value for LXYL-P1‒1V91S/I368E is 0.20 mM, and the kcat/Km for LXYL-P1‒1V91S/I368E is still 10.30 mM-1 s-1. Thus, the catalytic efficiency of LXYL-P1-1V91S/I368E was 2.5 and 1.8-fold higher than that of LXYL-P1-1 and LXYL-P1-1I368E, respectively.

Point 8: Line 181 Have you considered the LXYL-P1‒1A72T mutation?

Response 8: We have conducted the LXYL-P1‒1A72T mutation previously and the datum has been published. The mutant LXYL-P1-1A72T also exhibited the similar β-xylosidase and β-glucosidase activities compared with the high-active LXYL-P1-2, but were still lower than those of the mutant LXYL-P1-1V91S. Thus, we investigated that the enzyme activities of double mutant V91S/I368E in this study. As we mentioned in Conclusion part, the other combinatorial mutations (including A72T/I368E, A72T/I368T and V91S/I368T), may be conducted to find more active mutants of LXYL-P1-1.

Point 9: Line 191  Transformation with what? Should that “with” be there?

Response 9: We have deleted the word “with” in our revised manuscript.

By the way, we also revised the whole munuscript according to the attached pdf file “Molecules 2018 Manuscript Review”. The changes in the new version have been highlighted in red color.

Reviewer 2 Report

The manuscript „Site-directed mutagenesis of a beta-glycoside hydrolase from Lentinula edodes“ submitted by Jing-jing Chen and colleagues describes the biochemical characterization of a mutant and a double mutant of the enzyme LXYL-P1-1. Such enzymes are of tremendous interest due to their ability to be applicable in the production of the prominent anti-cancer drug Taxol.

Major concern:

The paper only contains a small dataset and is part of a series of papers from the same group including: Wang and Zhu 2013 Mycosystema; Cheng at al. 2013 MCP; Liu et al. 2016 Sci. Rep.; Liu and Zhu 2015 JIMB; Chen et al. 2017 Molecules. However, the data are relevant and worth to be published, but I would recommend to substantially edit the manuscript to come up with a short communication that may also be accepted for publication in “Molecules” (Instructions to Authors: Authors should not unnecessarily divide their work into several related manuscripts, although Short Communications of preliminary, but significant, results will be considered.).

Specific points:

In case of publication as a short communication, Figures 1 and 2 and Table 2 should be given as supplementary material.

Results and Discussion section only contains results so far. There are no data discussed and only a single paper is cited in the very first sentence. Some parts of the introduction describing other related enzymes may be included in the discussion section and the conclusion can be shortened, because at this moment it is mainly a repetition of the results section in condensed form (beside the very last sentence).

There are several data cited as “in press” (e.g. Line 72, 148, 156). A triple mutant has been generated and described in a separated publication probably showing identical experiments and comparable results. It is a pity that the data from this manuscript were not included into the description of LXYL-P1-1 triple mutant for direct comparison reasons. However, since the paper is “in press” it must be properly cited in the manuscript.

Minor points

Line 1 – Glycoside NOT glycoside

Line 2 – edodes NOT Edodes

Line 48 – subtilis NOT Subtilis

Line 50 – mannanase NOT mannase AND anserina NOT anserine

Line 87 and 91 – (Figure NOT ( Figure

Line 93 and 109 and 113 – xylosidase NOT Xylosidase

Lines 98-107 – Grouped style

Line 140 – activities and even NOT activities even

Line 152 – is a nonpolar NOT is an nonpolar

Line 153 – is a polar NOT is an polar AND acidic amino NOT acid amino

Line 154 – stability NOT stabilization

Figures 4A and 4B are not cited in the text

Line 188 – strains NOT strains

Line 191 – pastoris NOT Pastoris

Line 213 – H in BamHI not in italics

Headers 4.3, 4.4, 4.5, 4.6, 4.7 in “headline style”, but should be given in “sentence style” (see all other titles)

Line 220 – P. pastoris AND lxyl-p1-1 in italics

Line 222 – SacI  and P. pastoris in italics

Line 230 – lxyl-p1-1 in italics

References should be given in consistent form in “sentence style”. References 11 and 20 are given in “headline style”

Author Response

Response to Reviewer 2 Comments

Point 1:

Specific points:

In case of publication as a short communication, Figures 1 and 2 and Table 2 should be given as supplementary material.

Results and Discussion section only contains results so far. There are no data discussed and only a single paper is cited in the very first sentence. Some parts of the introduction describing other related enzymes may be included in the discussion section and the conclusion can be shortened, because at this moment it is mainly a repetition of the results section in condensed form (beside the very last sentence).

There are several data cited as “in press” (e.g. Line 72, 148, 156). A triple mutant has been generated and described in a separated publication probably showing identical experiments and comparable results. It is a pity that the data from this manuscript were not included into the description of LXYL-P1-1 triple mutant for direct comparison reasons. However, since the paper is “in press” it must be properly cited in the manuscript.

Response 1: Thank you for your quite professional comments. But, we have to say that we did not deliberately divide our work into several parts. Actually, apart from the preliminary results published on Mycosystema (2013), in one of our publications we carried out the directed evolution on LXYL-P1-2 and obtained a higher-active mutant LXYL-P1-2T368E. In our recent online publication, we carried out the site-directed mutagenesis from the low-active LXYL-P1-1 to the high-active LXYL-P1-2, in which the mutants LXYL-P1-1A72T, LXYL-P1-1V91S, LXYL-P1-1I368T, LXYL-P1-1A72T/V91S, LXYL-P1-1 A72T/V91S/I368Tand other mutants were obtained. The present manuscript was based on the information from the two papers and tried to observe the influence of LXYL-P1-1I368E and LXYL-P1-1V91S/I368E on the enzyme activity. Therefore, we think that the present work is a relatively independent story, although it is simpler than the present online publication.

Following your comments, we have carefully revised our manuscript as possible as we can. We have added the kinetic analysis of LXYL-P1-1I368T against XDT in our revised manuscript, and added the discussion about the comparison of LXYL-P1-1I368E with other reported mutants previously. The Conclusion Part has also been shortened in the new version. The paper “in press” has been properly cited in the new version.

By the way, following the suggestions of another reviewer, we have combined the Figure 1 and Figure 2 to ease comparison among all of the recombinants. For the readers to quickly capture the information from this paper, we still hope the manuscript is published in a full article.

Point 2: Line 1 – Glycoside NOT glycoside

Response 2: It has been revised as “Glycoside”.

Point 3: Line 2 – edodes NOT Edodes

Response 3: It has been corrected.

Point 4: Line 48 – subtilis NOT Subtilis

Response 4: It has been corrected.

Point 5: Line 50 – mannanase NOT mannase AND anserina NOT anserine

Response 5: They have been corrected.

Point 6: Line 87 and 91 – (Figure NOT ( Figure

Response 6: We have removed the space.

Point 7: Line 93 and 109 and 113 – xylosidase NOT Xylosidase

Response 7: We have revised these as “xylosidase”.

Point 8: Lines 98-107 – Grouped style

Response 8: We have revised the style.

Point 9: Line 140 – activities and even NOT activities even

Response 9: We have revised this sentence.

Point 10: Line 152 – is a nonpolar NOT is an nonpolar

Response 10: We have revised this sentence.

Point 11: Line 153 – is a polar NOT is an polar AND acidic amino NOT acid amino

Response 11: We have revised this sentence.

Point 12: Line 154 – stability NOT stabilization

Response 12: We have revised this sentence.

Point 13: Figures 4A and 4B are not cited in the text

Response 13: We have cited the figures in the text Line 151 in our new version.

Point 14: Line 188 – strains NOT Strains

Response 14: It has been revised as “strains”.

Point 15: Line 191 – pastoris NOT Pastoris

Response 15: It has been revised.

Point 16: Line 213 – H in BamHI not in italics

Response 16: It has been revised.

Point 17: Headers 4.3, 4.4, 4.5, 4.6, 4.7 in “headline style”, but should be given in “sentence style” (see all other titles)

Response 17: We have revised the headline style of 4.3, 4.4, 4.5, 4.6, 4.7.

Point 18: Line 220 – P. pastoris AND lxyl-p1-1 in italics

Response 18: They have been corrected.

Point 19: Line 222 – SacI  and P. pastoris in italics

Response 19: They have been corrected.

Point 20: Line 230 – lxyl-p1-1 in italics

Response 20: It has been corrected.

Point 21: References should be given in consistent form in “sentence style”. References 11 and 20 are given in “headline style”

Response 21: We have revised the References style.

Again, thank you very much for your comments and help.

Reviewer 3 Report

This manuscript describes that the double site-directed mutation in LXYL-P1−1 (V91S/I368E) improved the β-xylosidase and β-glucosidase activities. Authors interested in the β-glycoside hydrolases (LXYL-P1−1 and LXYL-P1−2) to hydrolyze 7-β-xylosyl-10-deacetyltaxol (XDT) to 10-deacetyltaxol and prepared mutants enzymes to evaluated these activities. In this paper, two mutated LXYL-P1−1were constructed and showed higher enzyme activity than the native one.

 The binding affinity with double mutant LXYL-P1−1V91S/I368E dramatically increased and resulted in the high β-xylosidase and β-glucosidase activities. In addition, authors considered that the V91S and I368E 20 mutation might positively promote the interaction between enzyme and substrate through altering 21 the loop conformation near XDT and increasing the hydrogen bonds among Ser91, Trp301 and XDT by the molecular docking between the substrate and enzyme. This work is very interesting, but the data is not enough to be concluded that combination of I368E and V91S improved the enzyme activity and catalytic efficiency. Authors should prepare the mono mutated enzyme, LXYL-P1−1 (V91S) and compare the activity with LXYL-P1−1 (V91S/I368E) to reveal the role of amino acid (E) at 368 in the enzyme activity. Authors already prepared the mutant LXYL-P1−1 (I368T) to exhibit high activity to similar to LXYL-P1−2. Authors should discuss the comparison of the activity between LXYL-P1−1 (I368T) and LXYL-P1−1 (I368E). In addition, authors should prepare the double mutant LXYL-P1−1 (V91S/I368T) and compare of the activity with  LXYL-P1−1(V91S/I368E).

At this stage, it is not enough to publish in this journal.           

Author Response

Response to Reviewer 3 Comments

Point 1: Authors should prepare the mono mutated enzyme, LXYL-P1−1 (V91S) and compare the activity with LXYL-P1−1 (V91S/I368E) to reveal the role of amino acid (E) at 368 in the enzyme activity.

Response 1: Thank you for your comments. We are sorry for the unclear description in the original manuscript and it has being addressed in the Introduction of the revised version. Actually, the mutant LXYL-P1−1V91S has been prepared in our previous study that has been published online now, and the LXYL-P1-1V91S exhibited the similar β-xylosidase and β-glucosidase activities compared with the high-active LXYL-P1-2. In this study, the β-xylosidase and β-glucosidase activities of LXYL-P1-1V91S/I368E were 2.3- and 1.5-fold higher than those of LXYL-P1-2, which revealed the important role of the Glu368 on the enzyme activity.

Point 2: Authors already prepared the mutant LXYL-P1−1 (I368T) to exhibit high activity to similar to LXYL-P1−2. Authors should discuss the comparison of the activity between LXYL-P1−1 (I368T) and LXYL-P1−1 (I368E).

Response 2: In order to make the comparison of the activity between LXYL-P1-1I368T and LXYL-P1-1I368E, we have added the kinetic analysis of LXYL-P1-1I368T against XDT in our revised manuscript. The results showed that the β-xylosidase and β-glucosidase activities of LXYL-P1-1I368E against PNP-Xyl and PNP-Glc were lower than those of LXYL-P1-1I368T. However, the catalytic efficiency of LXYL-P1-1I368E was 1.3-fold higher than that of LXYL-P1-1I368T. The possible reason for the inconsistency may be the different efficiencies of mutant in catalyzing the substrates with different molecular sizes.

Point 3: In addition, authors should prepare the double mutant LXYL-P1−1 (V91S/I368T) and compare of the activity with LXYL-P1−1(V91S/I368E).

Response 3: Thank you for your suggestion. Construction of the double mutant LXYL-P1−1 (V91S/I368T) requires a long time. It may be conducted in the future and we have addressed it in the Conclusions and perspective parts.

Round  2

Reviewer 3 Report

Authors have almost made corrections to follow the referee’s comment and I recommend this manuscript for publication.